# Nurses’ Occupational Stress and Presenteeism: The Mediating Role of Public Service Motivation and the Moderating Role of Health

**DOI:** 10.3390/ijerph18073523

**Published:** 2021-03-29

**Authors:** Hairui Jiang, Huanhuan Jia, Jingru Zhang, Yingying Li, Fangying Song, Xihe Yu

**Affiliations:** Social Medicine and Health Service Management, School of Public Health, Jilin University, No. 1163 Xinmin Street, Changchun 130021, Jilin, China; jianghr19@mails.jlu.edu.cn (H.J.); hhjia20@mails.jlu.edu.cn (H.J.); jrzhang19@mails.jlu.edu.cn (J.Z.); lyy18@mails.jlu.edu.cn (Y.L.); songfy18@mails.jlu.edu.cn (F.S.)

**Keywords:** challenge stress, hindrance stress, presenteeism, public service motivation, health

## Abstract

Presenteeism not only poses an economic cost to organizations but also generates reduced work efficiency and quality. The purpose of this study was to examine the connections between occupational stress, public service motivation (PSM), health, and presenteeism. A total of 981 nurses from 109 public hospitals in Jilin Province in China were enrolled in our study. Model 5 in the PROCESS micro was employed in order to verify the mediating effect of PSM and the moderating effect of nurses’ health on the relationship between occupational stress and presenteeism, and simple slope analysis was used to further determine the moderating effect. Both challenge stress and hindrance stress had a positive impact on presenteeism among nurses. PSM is a mediating variable between occupational stress and presenteeism. Health moderates the path between challenge stress and presenteeism, with the association being significant for nurses with low levels of health. Future policy making should focus on preventing presenteeism by reducing excessive stress, enhancing PSM, and improving nurse health and wellness.

## 1. Introduction

Presenteeism is the loss of organizational productivity among workers who show up to work but are not fully productive due to health problems [1], or else the behavior of employees who go to work even though they are not feeling well and should take sick leave [2,3], which is the definition adopted by most organizational scholars [4]. Compared to absenteeism, presenteeism is a more serious cause of productivity loss [5]. According to a survey in Japan, the annual monetary value lost to absenteeism was $520 per person, while that of presenteeism was $3055 per person [6]. Presenteeism poses a relatively substantial economic threat to organizations, and can generate reduced work efficiency and downgraded productivity in the workforce. Evidence has indicated that, in the USA, the indirect cost of presenteeism has reached $36 billion nationally, with a mean productivity loss of 13.2% [7]. Similarly, chief nurses in China estimated an average decline in work efficiency of 21.01% due to presenteeism among their subordinates [8]. Furthermore, a study showed that, in addition to the productivity loss due to presenteeism, it also results in an 18% increase in the number of patient falls and medical errors [9].

Due to the features of nurse work, such as shiftwork, inflexible work schedules, prolonged working time, and heavy workloads [10,11,12], nurses face extremely high stress, which not only harms their health but also decreases their productivity and keeps them from performing their jobs effectively in the workplace [12,13]. Previous studies have suggested that occupational stress is a predictor of presenteeism [14,15,16], while stressors of different natures may have different effects on individuals [17]. Cavanaugh separated challenge and hindrance stress based on their natures. In his longitudinal study, challenge stress, such as job responsibility and chances for learning, was positively associated with job satisfaction, but negatively associated with job searching; meanwhile, hindrance stress, such as role ambiguity and job insecurity, showed the opposite trend. In our study, we also classify stress into challenge and hindrance stress. Nurses experience higher presenteeism than workers in other industries [18]. Based on these findings, we propose the first hypothesis that challenge stress and hindrance stress significantly predicts nurse presenteeism.

Additionally, occupational stress is also related to employee work motivation. Public service motivation (PSM) is an important intrinsic motivation [19] defined by Perry and Wise as “an individual’s predisposition to respond to motives grounded primarily or uniquely in public institutions and organizations” [20]. When faced with occupational stress, employees’ motivation to engage in their work and to serve others and society might be influenced, potentially leading to a loss in productivity or presenteeism. Lepine’s study indicated that stressors account for 6% of the variance in motivation [21]. De Simone reported that PSM is positively correlated with engagement because, within public sectors, jobs are public-service oriented, which leads employees with a higher level of enthusiasm and dedication to work harder [22], thereby reducing the probability of presenteeism to some extent. There is still limited investigation on the effect of PSM on presenteeism caused by stress, especially among Chinese nurses in public hospitals; thus, it is necessary to carry out related research. Our second hypothesis is that nurse PSM mediates the link between occupational stress and presenteeism.

In view of previous studies, the relationship between health and presenteeism has been widely studied [23,24,25]. Specifically, the prevalence of presenteeism is higher in employees with higher health risk levels and a higher number of health risks, including physical inactivity and a higher body mass index [24]. The model of the mechanism linking health, productivity, and profit proposed by O’Donnell shows that health improvement brings an improved ability to work physically and emotionally, which can reduce absenteeism and presenteeism in return [25]. Excessive occupational stress has deleterious effects on mental and physical health [26,27], causing nurses to have fewer resources to engage in their work, resulting in presenteeism. In other words, the improvement of health levels can be recommended as a protective factor against the effect of occupational stress on presenteeism. Consequently, the third hypothesis of this study is that nurses’ health moderates the relationship between occupational stress and presenteeism.

In summary, we attempted to determine the mediating effect of PSM and the moderating effect of health between occupational stress and presenteeism among nurses in public hospitals in China. The theoretical model of this study is shown in Figure 1.

## 2. Material and Methods

### 2.1. Participants

The time span of this cross-sectional study was from 2 January to 15 January 2020. All public hospitals in Jilin Province were first stratified by region, and then local public hospitals were randomly selected according to number and size. Two county-level public hospitals, a public general hospital, and a public traditional Chinese medicine hospital were chosen from each county in Jilin Province, and 25% urban public hospitals were chosen from each city in Jilin Province in our study. In general, by using stratified sampling, 109 public hospitals, including 29 public hospitals at the city level and 80 public hospitals at the county level, were chosen in the research. Approximately 10 nurses were drawn from each hospital by convenience sampling. The inclusion criteria for the participants of the study were: in-service nurses aged between 18 to 60, nurses who could complete the questionnaire on their own, and nurses who agreed to participate in the study. The exclusion criteria for participants were: nurses on duty; nurses who were unwillingly to participate in the research. A total of 1052 in-service nurses, from 109 public hospitals in Jilin Province, China, took part in in our study. Excluding questionnaires with missing information, 981 nurses’ questionnaires were valid, and the validity rate was 93.25%.

### 2.2. Ethical Consideration

We obtained the approval of the Ethics Committee of the School of Public Health, Jilin University (No.20191203). With permission from each hospital, two or three medical staff members helped us to contact nurses from various departments, and to coordinate the survey. After asking all respondents to participate in the survey in a conference room in the hospital, we collected the information by using a structured questionnaire in the course of the survey. Prior to the survey, all of the subjects involved were informed of the objective of the study, and then the surveys were administered anonymously. In order to ensure quality and completeness, the submitted questionnaires were carefully reviewed by well-trained supervisors from the School of Public Health at Jilin University.

### 2.3. Measure Tools

Occupational stress was assessed by the challenge and hindrance-related self-reported stress (C-HSS) developed by Cavanaugh [28] and translated into a Chinese version by Yi [29]. The scale includes two dimensions, six items for challenge stress and five items for hindrance stress, for a total of eleven items. Each item consisted of five options, and the answers were “no stress”, “slightly stressed”, “stressed”, “very stressed”, and “extremely stressed”, with a score of 1, 2, 3, 4, and 5. The higher the score, the higher the stress. The C-HSS scale has been validated and shown to have good reliability in the Chinese context [30]. The Cronbach’s α coefficients of challenge stress and hindrance stress were 0.92 and 0.83, respectively.

The Stanford Presenteeism Scale-6 (SPS-6) [31] was used to measure presenteeism, which is a self-reported questionnaire including six items. The SPS-6 can be applied to study attendance among Chinese occupation groups with high reliability and validity [32]. The items “Despite having health problems, I was able to finish hard tasks at my work” and “At work, I was able to focus on achieving my goals despite my health problems” were reverse scored. Each item consisted of five options, which ranged from 1 to 5. Higher scores indicate higher presenteeism. In the present study, the Cronbach’s α coefficient of the SPS-6 scale was 0.80.

PSM was assessed by the Short-form PSM scale [33], which includes five items from Perry’s scale across four subdimensions [34]. The four subdimensions include social justice, self-sacrifice, commitment to public interest, and compassion. Although the PSM scale included only five items, it has been verified as a reasonable short measure [35]. Each item was scored from 1 to 5. In the current research, the Cronbach’s α coefficient of the PSM scale was 0.84.

Health was measured by the 8-item Short-form Health Survey Scale (SF-8), which analyzes the same eight health domains [36] as the Short-form Health Survey 36 (SF-36). The SF-8 is composed of eight questions which measure each domain of the SF36 [37]. All of the items were answered by the respondents on a five-point Likert scale ranging from 1 to 5. In the data collation phase, each item score was inverted so that higher values represented better health status. The Cronbach’s α coefficient was estimated to be 0.89, which indicates the high reliability of our research.

In addition, we also collected participants’ other information, including gender, age, marital status, education level, working years, and working hours per day.

### 2.4. Data Analysis

EpiData (Version 3.1, Epidata Association, Odense, Denmark) was used to input the collected data, and the PROCESS micro for SPSS (Version 24.0, IBM Corporation, Armonk, NY, USA) developed by Hayes [38] was utilized to analyze the data. First, we calculated the statistical descriptive information and the correlation matrix. Secondly, a moderated mediation effect was established using Model 5 in the PROCESS micro under conditions controlling for other variables. In addition, simple slope analysis was used to further determine the moderating effect. Both the mediating and moderating effects were tested by estimating the 95% confidence interval (CI) with 5000 bootstrap samples [39]. Confidence intervals that do not include zero reveal that they are statistically significant. All of the continuous variables in the current study were standardized before data processing.

## 3. Results

### 3.1. Participant Characteristics

Table 1 showed characteristics of the samples. In our samples, almost all of the participants were female (98.88%). The average age of the participants was 35.40 ± 8.31 years; 778 (79.31%) were married, 174 (17.74%) were unmarried, and 29 (2.96%) were divorced or widowed. In terms of education, 529 (53.92%) possessed a bachelor’s degree, and 27(2.75%) possessed a postgraduate’s degree. The average length of service was 12.35 ± 8.89 years. In addition, 162 (16.51%) worked less than 8 h a day, 661 (67.38%) worked 8 to 9 h a day, 101 (10.30%) worked 9 to 10 h a day, and the rest (5.81%) worked more than 10 h a day.

### 3.2. Correlation Matrix

Table 2 demonstrates the means, standard deviations, and correlation matrix for the main study variables. Both challenge stress and hindrance stress were significantly positively correlated with presenteeism (r = 0.34, *p* < 0.01; r = 0.48, *p* < 0.01), but were negatively correlated with health (r = −0.49, *p* < 0.01; r = −0.49, *p* < 0.01) and public service motivation (r = −0.29, *p* < 0.01; r = −0.34, *p* < 0.01). In addition, better health status (r = −0.52, *p* < 0.01) and higher public service motivation (r = −0.42, *p* < 0.01) were significantly negatively related to higher presenteeism.

### 3.3. Testing for the Moderated Mediation Model

Controlling for gender, age, marital status, education level, working years and working hours per day, model 5 in the PROCESS micro [38] was employed to verify the mediating effect of PSM and the moderating effect of nurse health on the relationship between occupational stress and presenteeism. As shown in Table 3, the results of Equation (1) showed that challenge stress had a negative effect on PSM (*β* = −0.31, *p* < 0.001). Equation (2) showed that challenge stress was significantly positively related to presenteeism (*β* = 0.07, *p* < 0.05), and that PSM was significantly negatively related to presenteeism (*β* = −0.27, *p* < 0.001).The 95% confidence interval did not contain 0, which indicates that the indirect effect of challenge stress on presenteeism was significant (bootstrap 95% CI = [0.06,0.11], *p* < 0.05). In addition, the interaction of challenge stress and health had a significantly negative predictive effect on presenteeism (*β* = −0.07, *p* < 0.01), suggesting that the mediating effect of PSM on the relationship between challenge stress and presenteeism tended to decrease with the improvement of health. Similarly, Equation (3) showed that hindrance stress was also significantly negatively associated with PSM (*β* = −0.35, *p* < 0.001), Equation (4) showed that hindrance stress had a significant positive effect on presenteeism (*β* = 0.21, *p* < 0.001), and that higher levels of PSM were related to lower levels of presenteeism (*β* = −0.23, *p* < 0.001). The indirect effect of hindrance stress on presenteeism was significant (bootstrap 95% CI = [0.05,0.10], *p* < 0.05), which meant that PSM acted as a mediating variable between the association of hindrance stress and presenteeism. At this point, the first and the second hypotheses held. The interaction of hindrance stress and health was non-significantly associated with presenteeism (*β* = −0.04, *p* > 0.05). The 95% confidence interval contained 0, indicating that health could not moderate the indirect associations between hindrance stress and presenteeism. Therefore, the third hypothesis partially held.

### 3.4. Simple Slope Analysis

Health was divided into a high level (84th percentiles) and a low level (16th percentiles), and we used simple slope analysis to further determine the moderating effect of challenge stress on presenteeism through PSM. As displayed in Figure 2, the simple slope analysis showed that, for nurses with low levels of health, lower levels of challenge stress were significantly associated with higher levels of presenteeism (95% CI = [0.01,0.07], *p* < 0.05). For nurses with high health levels, challenge stress did not have a significant effect on presenteeism through PSM. For nurses with high levels of health, the moderating effect of health was nonsignificant (95% CI = [−0.08,0.08], *p* > 0.05).

## 4. Discussion

The present study examined 981 nurses from Jilin Province in order to investigate the effect of occupational stress on presenteeism through a moderated mediation model, which was used to test the mediating role of PSM and the moderating role of health among nurses, which enriches the field of presenteeism research. As we expected, in the theoretical model, both challenge stress and hindrance stress positively predicted presenteeism. There is no doubt that hindrance stress includes stressful demands, which thwarts personal growth and goal achievement [28]. Our results are consistent with those of a previous study, which reported that hindrance stress significantly increases presenteeism [30]. Although challenge stress is viewed as an obstacle that can be overcome in order to learn and develop [28], this does not mean that the more challenge stress there is, the better. ‘Good stress’ has been characterized as stress that is not too high, or as stress on the upward part of the inverted U-shaped relationship between stress and performance [21]. Therefore, there may be a critical point in the influence of challenge stress on presenteeism. If this boundary is exceeded, the positive effect of challenge stress is weakened, and then presenteeism would increase. Nurses need to attend to patients and address patients’ existing or potential health problems. In addition, they also must deal with a complex work environment and manage personal relationships with fellows, doctors, and patients. Because of the complexity of the health system, nurses often experience tremendous stress, which inevitably leads to presenteeism.

The results of the correlation matrix showed that challenge stress and hindrance stress were both negatively related to PSM, which was consistent with the findings of some previous studies. Liu found that individuals with high levels of PSM easily handle work-related stressors better [40]. In another study, hindrance stress was also negatively associated with PSM, because hindrance stress adversely affected employees’ psychological condition [41]. Moreover, the positive effect of challenge stress was attenuated when nurses were under high stress and strains from work in the long term, and the negative effects of challenge stress were magnified; therefore, challenge stress can impair nurses’ passion and motivation for work. We also concluded that PSM had a negative effect on presenteeism, which is in line with Gross’s research [42], which indicated that compassion, one of the dimensions of PSM, is linked to increased presenteeism. Compared with private-sector workers, employees in the public sector value intrinsic rewards more, and obtain a feeling of accomplishment [43]. PSM motivates employees to work hard and reduces presenteeism. In addition, our results indicated that PSM mediates the relationship between occupational stress and presenteeism. According to cognitive interaction theory [44], when perceiving potential stress from work, nurses consume more psychological resources, including PSM, in order to overcome stress. However, when the loss of psychological resources and return are out of balance, negative emotions increase, and the enthusiasm of individuals to work decreases. Finally, it leads to job burnout, poor performance, and presenteeism.

According to the job demands–resources model [45], challenge and hindrance stress is one of the aspects of jobs that requires sustained effort, depletes energy, and impairs health, which inevitably leads to the excessive consumption and loss of coping resources. Our study showed that nurses’ health moderated the path between challenge stress and presenteeism, and the moderating effect was significant only in nurses with low levels of health. In other words, nurses’ low health status strengthens the negative effect of challenge stress on presenteeism. When nurses are in poor health, and at the same time need to cope with challenge stress, they appear to underperform, reduce work efficiency, and eventually fall into presenteeism conditions. However, for nurses with high levels of health, the effect of challenge stress on presenteeism was nonsignificant, which indicated that high levels of health suppressed the promoting effect of challenge stress on presenteeism. Consistent with many studies, employees in poorer health displayed higher presenteeism [23,24,46]. Besides this, health didn’t moderate the path between hindrance stress and presenteeism. The reason may be related to hindrance stressors, such as bureaucratic procedures and role conflicts, most of which cannot be overcome by the improvement of health, and are determined by organizational structure and organizational order [47]. In addition, compared with challenge stress, hindrance stress had a more obvious negative effect on presenteeism in our study, which may have resulted in the positive effect of health on presenteeism being offset.

Compared with workers in other sectors, employees within health services are more likely to experience presenteeism [2,48], especially nurses, who have a high prevalence of presenteeism [49]. Therefore, it is vital to prevent presenteeism and improve work efficiency. The present study discussed the mechanism between occupational stress and presenteeism. Moreover, we considered the role of PSM and the role of health in the relationship between occupational stress and presenteeism, which makes the association between them more specific. Our findings provide important implications for managers to prevent presenteeism among nurses in a focused and targeted manner. They should reduce hindrance stress by cutting down on red tape and establishing a fair organizational political system while controlling overloaded challenge stress by defining job duties and job responsibility. More attention should be paid to nurses’ health status. Hospital administrators should give enough time and vocational help for nurses in poor health to recover from their illness. In addition, it is necessary to establish a good organizational and working atmosphere, and to properly enforce the reward and punishment system in order to motivate PSM.

There are some limitations to this study. First, because of the lack of longitudinal tracking data, it is difficult to explain the causality between the variables. Longitudinal research designs should be given more attention in future studies. Second, the participants in this study were from public hospitals in Jilin Province, China, and the results may not hold nationally due to the source of the samples. Third, the measurement of the variables was based on subjective judgment. Not only subjective judgment but also subject data should be considered in future research. Finally, in addition to the mediating role of PSM and the moderating role of health, there may be some other mediating and moderating variables that also influence the relationship between occupational stress and presenteeism. Thus, more comprehensive and complex models should be constructed in order to further explore the mechanism of the action of stress on presenteeism.

## 5. Conclusions

In summary, after controlling for gender, age, marital status, education level, working years, and working hours per day, both challenge stress and hindrance stress are predictive of presenteeism among nurses. In addition, PSM is a mediating variable between occupational stress and presenteeism. Furthermore, health moderates the path between challenge stress and presenteeism, with the association being significant only for nurses with low levels of health. Future policy-making should focus on preventing presenteeism by reducing stress, enhancing PSM, and improving nurses’ health and wellness.

## Figures and Tables

**Figure 1 ijerph-18-03523-f001:**
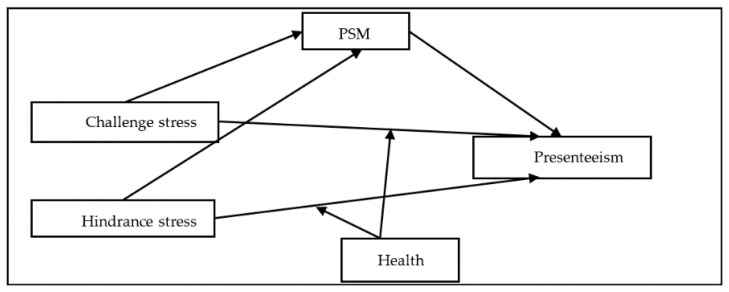
Hypothesis model: the association between occupational stress and presenteeism is mediated by Public motivation service (PSM), and is moderated by health.

**Figure 2 ijerph-18-03523-f002:**
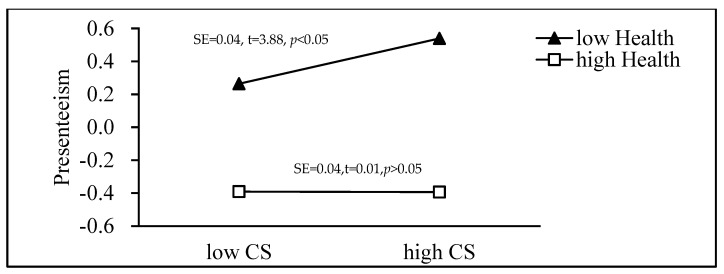
The moderating effect of health on the association between challenge stress and presenteeism through public service motivation (PSM).

**Table 1 ijerph-18-03523-t001:** Characteristics of the samples (*n* = 981).

Items	M ± SD/*n*(%)
Gender	
Male	11(1.12)
Female	970(98.88)
Age	35.40 ± 8.31
Marital status	
Unmarried	174(17.74)
Married	778(79.31)
Divorced or windowed	29(2.96)
Education Level	
Senior high school and below	425(43.32)
bachelor’s degree	529(53.92)
postgraduate	27(2.75)
Working years	12.35 ± 8.89
Working hours per day	
<8 h	162(16.51)
8~9 h	661(67.38)
9~10 h	101(10.30)
>10 h	57(5.81)

**Table 2 ijerph-18-03523-t002:** Correlation of the main study variables.

Variable	M ± SD	CS	HS	PSM	Health	Presenteeism
CS	12.93 ± 5.08	1	0.61 **	−0.29 **	−0.49 **	0.34 **
HS	8.89 ± 3.69		1	−0.34 **	−0.49 **	0.48 **
PSM	20.78 ± 3.39			1	0.321 **	−0.42 **
Health	33.93 ± 4.77				1	−0.52 **
Presenteeism	12.07 ± 4.80					1

Notes: *N* = 981; CS: challenge stress; HS: hindrance stress; PSM: public service motivation; ** *p* < 0.01.

**Table 3 ijerph-18-03523-t003:** Testing the mediated moderated effect on presenteeism.

Equation	Predictors	Outcome	*β*	t	95%CI		R^2^	F
					Lower	Upper		
Equation (1)	CS	PSM	−0.31	−9.96 ***	−0.38	−0.25	0.12	19.68 ***
Equation (2)	CS	Presenteeism	0.07	2.22 *	0.01	0.13	0.36	54.16 ***
	PSM		−0.27	−9.40 ***	−0.32	−0.21		
	Health		−0.37	−11.38 ***	−0.44	−0.31		
	CS × Health		−0.07	−2.86 **	−0.12	−0.02		
	Indirect effect		0.08		0.06	0.11		
Equation (3)	HS	PSM	−0.35	−11.74 ***	−0.41	−0.29	0.15	29.58 ***
Equation (4)	HS	Presenteeism	0.21	6.92 ***	0.15	0.27	0.39	69.92 ***
	PSM		−0.23	−8.27 ***	−0.28	−0.17		
	Health		−0.31	−9.96 ***	−0.37	−0.25		
	HS × Health		−0.04	−1.73	−0.08	0.01		
	Indirect effect		0.08		0.05	0.10		

Notes: CS: challenge stress, HS: hindrance stress, PSM: public service motivation; * *p* < 0.05; ** *p* < 0.01; *** *p* < 0.001.

## Data Availability

The data are not publicly available in order to protect the participants’ privacy.

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
