# Peer review of "Nurses’ Occupational Stress and Presenteeism: The Mediating Role of Public Service Motivation and the Moderating Role of Health"

_ijerph, 2021, doi:10.3390/ijerph18073523_

Round 1

Reviewer 1 Report

Minor correction needed before the publication.

Comments for authors:

Thank you for the opportunity to review the article entitled “Nurses’ Occupational stress and Presenteeism: The mediating role of Public Service Motivation and the moderating role of Health”.

Well-written paper. A few suggestions that need improvement before publication:

  1. Abstract should be prepared according to the guidelines for authors: “abstract should be a single paragraph and should follow the style of structured abstracts, but without headings”, please see: https://www.mdpi.com/journal/ijerph/instructions.
  2. Line 105-133: these data should be placed in the results section as a characteristic of the study group.
  3. In the methodology section, please provide information about the inclusion and exclusion criteria for participance of the study.
  4. Line 114-122 – Ethical consideration: Please provide the number and date of the Bioethics Committee.

Author Response

Response to Reviewer 1 Comments

Point 1: Abstract should be prepared according to the guidelines for authors: “abstract should be a single paragraph and should follow the style of structured abstracts, but without headings”, please see: https://www.mdpi.com/journal/ijerph/instructions.

Response 1: We have modified the Abstract according to the guidelines for authors.

Point 2: Line 105-133: these data should be placed in the results section as a characteristic of the study group.

Response 2: We have added a subsection with detailed demographics of study participants in our paper according to your suggestion, Please see Table 1.

Point 3: In the methodology section, please provide information about the inclusion and exclusion criteria for participance of the study.

Response 3: We have provided provide information about the inclusion and exclusion criteria for participance of the study. Please see Line 152 and 155.  “Inclusion criteria for participants of the study: In-service nurses aged between 18 to 60; nurses who can complete the questionnaire on their own, nurses who agree to participate in the study. Exclusion criteria for participants: nurses on leave or off-duty; nurses who are unwillingly to participate in the research.

Point 4:Line 114-122 – Ethical consideration: Please provide the number and date of the Bioethics Committee.

Response 4: We have provided the number and date of the Bioethics Committee. Please see Line 160 and 161. “We obtained the approval of the Ethics Committee of the School of Public Health, Jilin University(No.20191203).”

Reviewer 2 Report

In general this paper is interesting and well written.  The introduction, purpose of the study and hypotheses are clearly stated.  The discussion of the results is fine, and so is the conclusion.

However, I have a number of problems with the description of the methodology.  The document herewith presents a number of questions and clarifications which would need to be addressed to make this paper publishable.

Author Response

Point 1: How were the hospitals selected?  At line 100 it is stated that “a certain percentage of hospitals”; what percentage? The selection criteria are unclear.

Response 1: Dear reviewer, thanking you a lot for your suggestion. we selected two county-level public hospital from each county in Jilin Province and 25% urban public hospitals from each city in Jilin Province. Because the size and number of hospitals are mainly limited by the development level of the region, so this study used two sampling patterns in the county and city respectively. We clarify the section criteria in the paper. Please see line 116. Two county-level public hospitals, a public general hospital and a public traditional Chinese medicine hospital were chosen from each county in Jilin Province and 25% urban public hospitals were chosen from each city in Jilin Province in our study.

Point 2:The way data were collected are unclear.  In the methodology section, one gets the impression that the questionnaire was self-administered.  However, at line 294 the authors write that “the measurement of variables was 294 based on the subjective judgment and evaluation of the investigators”, which seem to imply that the nurses were actually “interviewed” by the investigators who then noted the answers on the questionnaire.  So how were the data collected?

Response 2: We really appreciate your suggestion. The sentencethe measurement of variables was based on the subjective judgment and evaluation of the investigators were replaced by the measurement of variables was based on the subjective judgment after careful consideration. We collected data through self-administered questionnaires.

Point 3: Do we understand that the participating nurses were actually gathered in a conference room?  During their work shift?  If that is the case, didn’t that have a negative effect on current work in the various hospital departments?

Response 3: Dear reviewer, in our study, the participating nurses were actually gathered in a conference room. Yet prior to survey, we got in touch with hospital coordinators and obtained the help from them. They helped us organize off-duty nurses into conference room to take part in our study. Besides, we supplemented the exclusion criteria for nurses in line 154. Exclusion criteria for participants: nurses on duty; nurses who are unwillingly to participate in the research.During the investigation, we tried to minimize the negative impact on current work in the various hospital departments.

Point4. And on average, how long did it take to complete the questionnaire?

Response 4: Nurses needed 8~15 minutes to complete the questionnaire.

Reviewer 3 Report

I’ve read with interest the paper from JIANG et al, reporting on the mediating role of Public Service Motivating role of Health in occupational stress and presenteeim in nurses from the Chinese province of Jilin.

Albeit of interest, the present version of this paper is affected by several shortcomings that, in my opinion, represent a significant barrier for an eventual publication.

More precisely:

  1. Major Concerns:
    1. The study population is not properly described; some snapshots are given in the 2.1 section, but information such as monthly income and detailed demographics are not properly reported, but only summarized. As all the analyses were adjusted by demographics, it is impossible to properly judge the results of the study. Authors should rewrite Results including a subsection with detailed demographics of study participants, summarizing such details in a specific table. Moreover, participation rate remains unclear: authors should explain whether all 981 nurses with completed data eventually participated to the survey. Moreover, do you have any info about the potential statistical power of this sample?
    2. Statistical analysis is improperly reported. Authors performed a series of regression analysis models that are not clearly described across the main text, in particular in terms of explanatory variables that were eventually included. Moreover, the results are not clearly reported: for example, in Table 1 Rows’ heading should be clarified.
    3. In simple slope analysis, health was dichotomized by mean value. This is quite uncommon as usually an approach by median value is preferred. Could you please explain your choice?
  2. Minor Concerns:
    1. Introduction is too long, some sections could be moved to the discussion, particularly when dealing with features of PSM that are not absolutely necessary to introduce this topic;

Author Response

Point 1. The study population is not properly described; some snapshots are given in the 2.1 section, but information such as monthly income and detailed demographics are not properly reported, but only summarized. As all the analyses were adjusted by demographics, it is impossible to properly judge the results of the study. Authors should rewrite Results including a subsection with detailed demographics of study participants, summarizing such details in a specific table. Moreover, participation rate remains unclear: authors should explain whether all 981 nurses with completed data eventually participated to the survey. Moreover, do you have any info about the potential statistical power of this sample

Response 1: Firstly, we have added a subsection with detailed demographics of study participants in our paper , Please see Table 1.

Secondly, we also clarify participation rate in the paper. A total of 1052 in-service nurses, from 109 public hospitals in Jilin Province, China, took participate in in our study. Excluding questionnaires with missing information, 981 nurses’ questionnaires were valid and validity rate was 93.25%.

Point 2. statistical analysis is improperly reported. Authors performed a series of regression analysis models that are not clearly described across the main text, in particular in terms of explanatory variables that were eventually included. Moreover, the results are not clearly reported: for example, in Table 1 Rows’ heading should be clarified.

Response 2: We deleted extra regression analysis models that are not clearly described across the main text, and we refined the result of Table 3. Besides, Table 1 Rows’ heading was clarified. Please see line 286.

Point 3.In simple slope analysis, health was dichotomized by mean value. This is quite uncommon as usually an approach by median value is preferred. Could you please explain your choice

Response 3:Dear reviewer, the median value is preferred because of the limitation of distribution of variable. In order to characterize moderating effect and conduct simple slope analysis, PROCESS micro for SPSS provides two ways for conditioning values divide into a high level and a low level. One is (-1SD,Mean,+1SD) and the other is (16th, 50th ,84th percentiles). It is more appropriate to adopt median. So we reconstructed the simple slope analysis in our paper.

Point 4. Introduction is too long, some sections could be moved to the discussion, particularly when dealing with features of PSM that are not absolutely necessary to introduce this topic;

Response 4:Dear reviewer, We abbreviated the introduction according to your suggestion, which was valuable to us. Thank you a lot.

Round 2

Reviewer 3 Report

Estimated Authors,

I've appreciated the efforts you paid in order to improve your paper: actually, the large majority of all my concerns been addressed, and the main shortcomings of the precedent version as well.

At the moment, I've only a couple of FORMAL suggestions, and namely:

1) Table 1, in order to avoid potential misunderstandings, I would change the row AGE and Working Years as follows --> 35.40(8.31) --> 35.40 ± 8.31; 12.35(8.89) --> 12.35 ± 8.89

2) headings of Table 2 remains unfixed, including variables 1,2,3,4,5,: please clarify

3) as Tables should be self-explanatory, please include either in Notes or in the label what the Model1,2,3,4 actually represent.

After such interventions, I think that the article may be published in IJERPH

Author Response

Ponit1: Table 1, in order to avoid potential misunderstandings, I would change the row AGE and Working Years as follows --> 35.40(8.31) --> 35.40 ± 8.31; 12.35(8.89) --> 12.35 ± 8.89

Response 1:Dear reviewer, we really appreciate your generous help. We have changed the row age and working years according to your suggestion.

Point 2: headings of Table 2 remains unfixed, including variables 1,2,3,4,5,: please clarify

Response 2: Dear reviewer, headings of Table 2 was modified .

point 3: as Tables should be self-explanatory, please include either in Notes or in the label what the Model1,2,3,4 actually represent.

Response 3: Dear reviewer,  in order to avoid misunderstanding, we have replaced Model 1,2,3,4, by Equation 1,2,3,4. And the meaning and  results  of each Equation was represented in the 3.3. 

Dear reviewer, thank you for your all suggestions which is of great valuable to us. 
